# Improving sentiment classification with task-specific data

## Abstract

Current state-of-the-art sentiment analysis techniques rely heavily on pre-trained word embeddings. However, the data used to train these embeddings normally comes from large, generic datasets, such as Wikipedia or GoogleNews, which may not include enough task-specific information to create reliable representations. This paper proposes a method to determine the subjectivity of a corpus using available tools and shows that word embeddings trained on task-specific corpora tend to outperform those trained on generic data. We then examine ways to combine information from generic and task-specific datasets and finally demonstrate that our method can work well for under-resourced languages.

## 1 Introduction

Sentiment analysis techniques requiring only word embeddings as features have proven to be successful (Socher et al., 2013; Tang et al., 2014; Iyyer et al., 2015). Research has also shown the importance of properly initializing the word embeddings, either through careful manipulation of the initialization of random vectors, or more commonly by pretraining the word embeddings (Kim, 2014). The content of the corpora used to train these word embeddings, however, has not been examined in depth.

Most research tends toward using larger and larger datasets in order to build high-quality word embeddings, with some studies using corpora on the magnitude of several billion tokens (Mikolov et al., 2013; Pennington et al., 2014). This amount of data, however, is not available in all languages, nor is it necessarily an optimal use of available re-sources. In fact, there is evidence suggesting that tailoring a dataset for a task can achieve better results with less data (Suster et al., 2016; Barnes et al., 2016).

The subjectivity or polarity content of a corpus could provide a good clue to the quality of word embeddings created for the task of sentiment analysis. Pang and Lee (2004) showed that using a pipeline technique where they first classified subjectivity before classifying polarity, they were able to achieve better results than performing classification without filtering subjective sentences. Therefore, we attempt to use subjectivity as a metric for the appropriateness of a corpus for our task.

In this work we aim to:

- discover if small task-specific datasets can provide better representations of words than large generic datasets for the task of sentiment analysis

- quantify the amount of subjectivity in the data in order to predict the appropriateness of a dataset for the task of sentiment analysis.

- discover the best way to combine this information for our task, given a large generic dataset and a smaller domain-specific dataset.

- extract the subjective information as a way of approximating a smaller task-specific dataset, given only large generic datasets,

As far as we know, this is the first work on the effects of the subjectivity and task-specificity of a corpus on the creation of word embeddings for the task of sentiment analysis. Given the importance of initialization and the wide-spread use of sentiment analysis techniques using word embeddings, the results are of importance to the community.

This paper is organized as follows. Section 2 discusses previous work on subjectivity, generic word embeddings and task-specific word embeddings. Section 3 outlines how to quantify the subjectivity of a corpus. Section 4 determines how subjectivity and task-specificity effects word embeddings, and how to combine information from different datasets in order to improve classification. In Section 5 we propose a method to extract task-specific information from generic corpora and, by using the techniques from Section 4.2, improve over our baseline. Finally, in Section 6 we show that this technique of concatenation representations works even better for under-resourced languages.

## 2 Related Work

### 2.1 Subjectivity as a filter for polarity

Subjectivity can be a clear indicator of whether a sentence or phrase contains an opinion. In the case of movie or product reviews, this is especially relevant as there are often sentences that seemingly contain sentiment information, yet do not give an opinion towards any relevant aspect: for example, "The main character tries to protect her good name." Here, 'good' gives no relevant sentiment information and could easily be found within a negative review.

Pang and Lee (2004) use a subjectivity classifier to extract relevant sentences before passing them on to a sentiment classifier. Reducing the amount of irrelevant or potentially misleading data presumably leads to an improvement in polarity classification.

This approach could potentially improve vector representations of words by providing only relevant examples to the word embedding algorithm.

### 2.2 Representing Words as Vectors

Representing words as feature vectors is an idea that has created interest for many years (Deerwester et al., 1990; Lund and Burgess, 1996). The simplest example is to represent a word as a one-hot vector the length of the vocabulary, where all entries are zeros except for the index of the word, which is a 1. However, this simple technique does not allow for generalization, as the words 'dog' and 'cat' are as different as 'dog' and 'dogs'.

Lower-dimensional dense vectors, often known as word embeddings, are able to avoid this problem to a degree. They do this by attempting to force similar words to occupy a similar area in vector space, given an objective function. The most successful of these applications to date is the Skip Gram with Negative Sampling (SGNS) algorithm (Mikolov et al., 2013). This algorithm applies PMI factorization over an implicit word/feature matrix (Levy and Goldberg, 2014).

Although generic word embeddings are often used as features for sentiment classifiers (Socher et al., 2013; Kim, 2014), there are good reasons to believe that the generality of these vectors may lead to inferior results. Most state-of-the-art sentiment classifiers are based on deep neural networks. With these architectures it is rather difficult, if not impossible, to find the global optimum and therefore, we must usually be content to find a local optimum. This fact also means that they are sensitive to the initialization of their word embeddings. For this reason, sentiment classifiers initialized with random vectors consistently underperform those initialized with pretrained vectors (Kim, 2014; Iyyer et al., 2015).

### 2.3 Task-specific Word Embeddings

Task-specific word embeddings are a further development of word embeddings. The purpose is to provide a representation that helps a classifier, often by removing or adding task-specific information.

These vectors can be trained as normal and then retrofitted to the task (Yu and Dredze, 2014; Xu et al., 2014; Faruqui et al., 2015). The advantage of this technique is that any kind of word embedding can be used, as the vectors are modified post training. Faruqui et al. (2015) propose a method to incorporate information from semantic lexicons (WordNet, FrameNet, and ParaphraseNet) into word embeddings.

Task-specific word embeddings can also be trained in a supervised manner, using the labels from annotated corpora (Maas et al., 2011; Tang et al., 2014, 2016) or task-specific dictionaries (Gouws and Søgaard, 2015). For the inclusion of sentiment information in word vectors, this is the most common approach.

Maas et al. (2011) build a probabilistic model, similar to LDA (Blei et al., 2003), and attempt to model both word relationships and sentiment at the same time. However, their sentiment vectors performed worse than generic word vectors without the sentiment information.

Tang et al. (2014) attempt to create sentiment-specific word vectors by modifying the feed-forward network architecture used by Collobert and Weston (2011) with a hinge-objective that is designed to maximize both the word representation and the model's ability to classify polarity. They collected 10 million distantly supervised tweets by searching for positive and negative emoticons. They trained a sentiment classifier and achieved state-of-the-art results on the SemEval-2013 shared task (Nakov et al., 2013).

Both of these techniques attempt to use large amounts of annotated data to extract sentiment information, but do not consider different sources of data or ways to combine them. However, it would be beneficial to examine more in depth the type of data that we use to create word vectors.

The work that most closely resembles our is that of Joshi et al. (2015). They explore the effects of using in-domain and generic datasets to create embeddings for e-commerce NER. Their work shows that training on smaller amounts of task-specific data is better than training on large amounts of generic data. Unlike our work, they do not look into ways of combining information from different sources.

## 3 Quantifying the amount of subjectivity in corpora

### 3.1 Datasets

Our first goal was to quantify the amount of subjective information contained in a corpus. We collected the following corpora, which we believe represent both generic datasets often used in the literature (Wikipedia, Europarl, Multi UN) as well as task-specific corpora (Amazon Movie Dataset). All corpora were sentence tokenized, word tokenized and lower-cased using Stanford Core NLP (Manning et al., 2014). We give the details of each corpus in the following subsections as well as in Table 1.

#### Wikipedia Corpus

After downloading a 2016 Wikipedia dump, we removed duplicates and markup using a freely available script[1].

---

[1] http://attardi.github.io/wikiextractor/

#### English Europarl

The English part of the Europarl v7 corpus[2] (Koehn, 2005) is composed of around 2 million sentences from the European Parliament.

#### Multi UN

MultiUN[3] is a corpus extracted from the official documents of the United Nations (Eisele and Chen, 2010). We use the English side of the English-Spanish corpus. It contains around 11 million sentences.

#### Amazon Movie Corpus

This dataset is composed of more than 7.5 million sentences taken from nearly 1.7 million reviews about movies (McAuley et al., 2015).

### 3.2 Method

In order to determine the amount of subjectivity contained in each corpus, we ran OpinionFinder on each of them. This is a system that was designed for subjectivity and polarity detection (Wilson et al., 2005). It has a high-precision and high-recall subjectivity classifier. The high-precision classifier uses a rule-based method, looking for well-established lexical clues. The high-recall classifier is a naive bayes classifier trained on the MPQA corpus. After running OpinionFinder on each corpus, we retrieved the results for high-precision subjectivity and high-recall subjectivity. In order to quantify the amount of subjectivity, we took the number of sentences reported as subjective and divide them by the total number of sentences in the corpus, (see Formula 1).

$$Subjectivity\ Score = \frac{|subjective\ sentences|}{|total\ sentences|} \tag{1}$$

This gives us an idea of the subjectivity content of one corpus compared to the others, although it is not a precise measurement. We did the same with high-precision and high-recall to get two scores for each corpus (see Table 1). As we expected, the Wikipedia corpus has the lowest scores for subjectivity. This is likely because of its encyclopedic format. The corpora with the highest subjectivity scores are the Amazon (high precision) and Europarl (high recall) corpora.

---

[2] http://www.statmt.org/europarl
[3] http://opus.lingfil.uu.se/

|  | Wiki | Multi UN | Europarl | Amazon |
|---|---|---|---|---|
| High recall subj. | 14.8% | 33% | 49% | 41.5% |
| High precision subj. | 2.8% | 5.1% | 6% | 9.3% |
| # Sentences | 118,859,889 | 11,350,967 | 1,965,734 | 7,540,838 |
| # Tokens | 2,099,860,107 | 322,056,452 | 54,474,544 | 162,390,631 |
| Avg. Sent Length | 17.7 | 28.4 | 27.7 | 21.5 |

Table 1: Statistics of corpora used to train word embeddings.

## 4 Experiments

### 4.1 Baseline experiment

We created 50- 100- and 300-dimensional word vectors for each corpus using the Skip-gram with Negative Sampling algorithm (Mikolov et al., 2013) with a window of 10 words, and 5 negative samples.

We tested the word vectors on the Rotten Tomatoes dataset (Pang and Lee, 2005), which contains 10662 annotated sentences. We train on 7996, leave 1066 as a development set and test on 1600.

In order to see how well these representations work on a different domain, we also tested them on the English OpeNER sentiment corpus (Agerri et al., 2013). Specifically, we take the subcorpus from the hotel domain. It contains 3709 opinion phrases annotated for opinion holder, opinion target and four levels of sentiment (Strong Positive, Positive, Negative and Strong Negative). Each training example is a tuple of the opinion holder, opinion target and sentiment phrase, such as ("we", "hotel", "didn't like at all"). We train on 2780, leave 186 as a development set and test on 743. This dataset has a class imbalance, which means that macro f1 is a more appropriate metric.

Following the work of Iyyer et al. (2015), we use a Deep Averaging Network to perform sentiment classification. This neural bag-of-words model does not take the order of the words into consideration, but has proven to give results similar to more sophisticated approaches. Our datasets do not allow us to train on labeled phrases. Also the small size of the training data, especially in the OpeNER dataset, means that updating the word vectors while training the classifier actually hurts classification when using our task-specific vectors. Therefore, we do not update while training.

For each sentence or phrase in the datasets, we create a training example. A training example is the average of the word embeddings for each word in the sentence or phrase. If a word is not found in our word embedding model, it is replaced with a generic 'unknown' vector, which is randomly initialized. The results are shown in Figures 1 and 2.

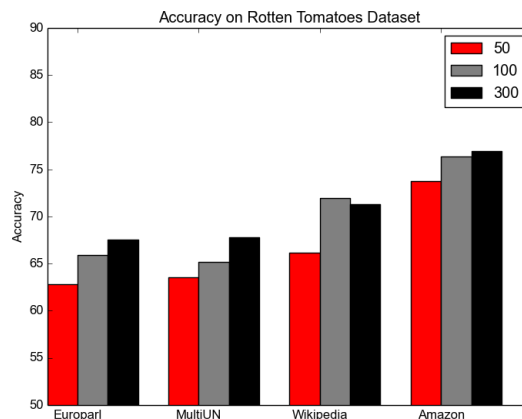

Figure 1: Accuracy of sentiment classifiers on Rotten Tomatoes dataset

As we can see, the embeddings trained on the much smaller Amazon corpus give consistently better results than the Wikipedia corpus on the RT dataset. However, the vectors trained on Wikipedia perform quite well and even outperform their Amazon trained counterparts on the OpeNER dataset. This is likely due to the fact that the domain for the OpeNER dataset is different (Hotel reviews) compared to the RT dataset (Movie Reviews).

### 4.2 Combining information from different sources

Given that vectors trained on task-specific data regularly outperform those trained on much larger generic datasets, we should ask ourselves if it is possible to combine both of these datasets to get even better results. We compare three methods:

1. Appending the task-specific dataset to the generic dataset and training one representation

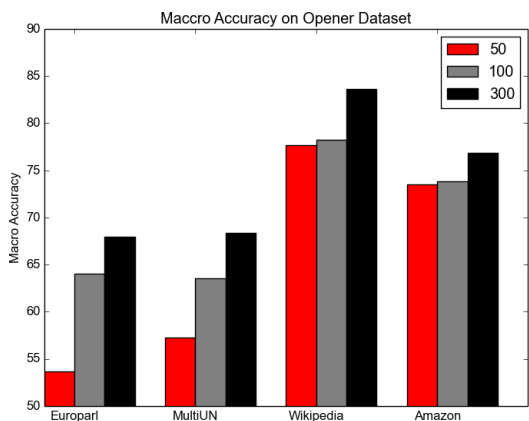

Figure 2: Accuracy of sentiment classifiers on OpeNER dataset

2. Splicing the task-specific dataset into the generic dataset and training one representation

3. Training two separate representations and combining them through vector concatenation.

Appending involves simply adding the task-specific Amazon Movie corpus to the Wikpedia corpus and training a single set of vectors on this data. This should improve the representations of those words that are important for the task, as they will receive more positive examples while training the vectors.

Splicing is similar to appending, but instead of adding the Amazon sentences, we splice them into the Wikipedia corpus. We alternate sentences from the two corpora until we have exhausted all of the Amazon sentences, after which we add the remaining Wikipedia sentences. As above, we train a single set of vectors.

For concatenation, we train one set of vectors on Wikipedia and a second on the Amazon Movie Corpus separately. Given two $m$ dimensional vectors, we concatenate them, resulting in an $m + m$ dimensional vector, in order to provide both views to the sentiment classifier. For a fair comparison, we concatenate vectors that are half as large as those of the other techniques. This technique may provide better representations for words that often appear in one corpus but not in the other.

Note that the embeddings in these three techniques were trained on exactly the same data, the only difference being the method of combining

this data. Our baseline vectors are trained only on the Wikipedia corpus. We also test the 50 dimensional sentiment embeddings ($SSWE_u$) provided by Tang et. al (2014) (see Section 2.3). The results are shown in Table 2.

We can see that including task-specific data with any of the techniques tends to improve over the baseline. This in itself is not surprising. However, it is important to notice that the concatenation of two vectors trained on the two datasets separately always improve the macro f1 score.

### 4.3 How much task-specific data is needed in order to see benefits

Given that we can combine generic and task-specific data to improve our feature space, we would like to know how much task-specific data is necessary to see these improvements. Therefore, we perform experiments on the RT dataset with increasing amounts of task-specific data. The results are shown in Figure 3.

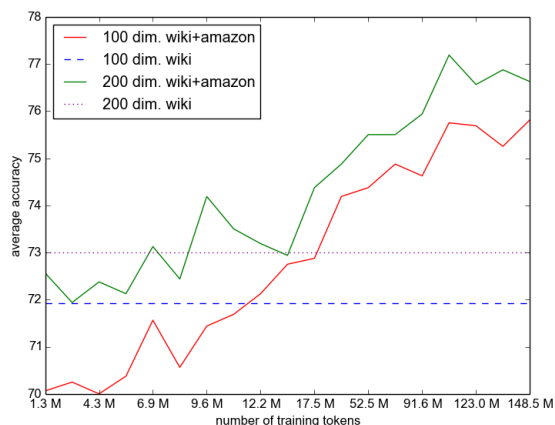

Figure 3: The benefits of concatenating wiki and amazon vectors plotted as a variable of amazon tokens used

As we can see from the results, we see improvements over the baseline after 10-12 million tokens. This could potentially benefit low-resource languages which often have access only to Wikipedia pages and little task-specific data.

## 5 Extracting task-specific information from generic datasets

Finally, we ask if there is a way to approximate this task-specific data given only large, generic corpora. This again is a situation that is very common among under-resourced languages.

| | | RT | | | | OpeNER | | | |
|---|---|---|---|---|---|---|---|---|---|
| | | 50 | 100 | 200 | 300 | 50 | 100 | 200 | 300 |
| Wikipedia Baseline | acc | 66.2 | 71.9 | 71.4 | 71.3 | 77.6 | **78.2** | 82.1 | **83.6** |
| | f1 | 59.8 | 72.4 | 67.4 | 67.5 | 37.2 | 36.7 | 39.8 | 40.6 |
| SSWE$_u$ | acc | 66.3 | – | – | – | **81** | – | – | – |
| | f1 | 56.5 | – | – | – | 39.4 | – | – | – |
| Amazon | acc | 73.8 | **76.3** | **76.7** | **76.9** | 73.5 | 73.8 | 77.8 | 76.9 |
| | f1 | 69.9 | 74.7 | 74.6 | 75.3 | 34.9 | 33.6 | 37.4 | 38 |
| Appended | acc | 69.4 | 73.8 | 74.5 | 74.3 | 77.6 | 76.5 | 82.7 | 78.4 |
| | f1 | 65.1 | 75 | 72.5 | 72.8 | 36.9 | 35.8 | 40.5 | 36.4 |
| Spliced | acc | 70.8 | 71.2 | 73.1 | 73.6 | 77.4 | 73.9 | **84.6** | 82.7 |
| | f1 | 62.5 | 73.4 | 71 | 70 | 37.2 | 32.9 | 41.4 | 39.9 |
| Amazon + Wiki | acc | **74.4** | 75.2 | 75.9 | 75.9 | 64.2 | 65.7 | 69.4 | 72.3 |
| | f1 | **72.5** | 75 | **78** | **77.7** | **57.8** | **62.4** | **63.8** | **68** |

Table 2: Accuracy and F1 of sentiment classifier with embeddings trained on Wikipedia and Amazon corpora with different techniques. For the OpeNER dataset, accuracy is the macro average. **Bold** indicates the highest performance and underline indicates those that beat the generic baseline.

fore, we extract the sentences labeled as subjective from the Wikipedia, Europarl and Multi UN corpora, referred to from here on as subj-Wikipedia, subj-Europarl, subj-MultiUN. We train word embeddings on each of these and test them as in Section 4.2. We also report macro-averaged f1 scores. Results are shown in Table 3.

We can see that by filtering the corpora for subjectivity, we lose a degree of accuracy, but gain in f1. The poor f1 scores of the Wikipedia baseline on the OpeNER dataset are due to the fact that the classifier nearly always chooses the two majority classes and completely ignores the others. Including more subjective information seems to ameliorate this situation. The improvements in f1 also coincide with the subjectivity level predicted by our method. This suggests that it is possible to predict the amount of improvement that you can gain by extracting subjective data from a generic dataset.

## 5.1 Lexical Overlap

Given that the classification error is still relatively large, our hypothesis was that many words which are important for classification are not found in the corpora used to train the word embeddings. In order to test this possibility, we found the lexical overlap in vocabulary between the corpora used to train the word embeddings and the RT and OpeNER corpora. Let $V_{tr}$ be the set of words in a training corpus and $V_{test}$ be the set of words in each test corpus $test$.

$$LexicalOverlap = \frac{|V_{test}| \cap |V_{tr}|}{|V_{test}|} \quad (2)$$

If the lexical overlap of the corpora used to train our representations (Wikipedia, Europarl, Multiun) and the datasets we use to test our classifier (RT, OpeNER) is small, this will negatively affect the results of the classifier, as it will lack necessary information during both training and testing. We also test for lexical diversity of the filtered subjectivity training corpora as a measure of vocabulary. We use the Average Type Token Ratio (**ATTR**) as a measure. Let $k$ be a corpus, $n_k$ be a subset of $n$ tokens of $k$, $V_n$ = the vocabulary of $n$, and $T_n$ = the tokens of $n$.

$$\text{Average Type Token Ratio} = \frac{1}{n} * \sum_{n=1}^{N} \frac{|V_n|}{|T_n|} \quad (3)$$

We set $n$ to 1000. The results are shown in Table 4.

As we can see in Table 4, the number of words that do not have a vector representation in the full Wikipedia model is much smaller than all other models. Therefore, the Wikipedia word embeddings have a larger vocabulary yet they do not always have a representation that is helpful for the task. The more subjective datasets may lack some vocabulary, but it seems that the representations that they have are able to overcome the sparsity.

|  |  | RT | | | | OpeNER | | | |
|---|---|---|---|---|---|---|---|---|---|
|  |  | 50 | 100 | 200 | 300 | 50 | 100 | 200 | 300 |
| Wikipedia Baseline | acc | 66.2 | **71.9** | **71.4** | 71.3 | **77.6** | **78.2** | **82.1** | **83.6** |
|  | f1 | 59.8 | 72.4 | 67.4 | 67.5 | 37.2 | 36.7 | 39.8 | 40.6 |
| subj-Wiki + Wiki | acc | 66.9 | 69.1 | 70.3 | 72.4 | 62.1 | 67.6 | 67.8 | 70.3 |
|  | f1 | 67.5 | 71.8 | 74.6 | 75.5 | 51.2 | 58.3 | 57.9 | 60.8 |
| subj-Multiun + Wiki | acc | **68.4** | 69.6 | 69.3 | 73.4 | 65.1 | 68.3 | 68.2 | 71 |
|  | f1 | 69 | 73.2 | 74 | 75.6 | **60.2** | **60.7** | 57.7 | 62.8 |
| subj-Europarl + Wiki | acc | 68.2 | 68.4 | 71.3 | **74.1** | 63.7 | 68.3 | 69.6 | 71.7 |
|  | f1 | **69.7** | **73.7** | **75.5** | **76.2** | 56.9 | 59.5 | **66.3** | **63.4** |

Table 3: Accuracy and F1 scores of sentiment classifier trained on concatenation of filtered subjectivity vectors and wikipedia vectors. **Bold** indicates the highest performance and underline indicates those that beat the Wikipedia baseline.

|  | Wiki | Amazon | subj-Wiki | subj-MulUN | subj-Euro |
|---|---|---|---|---|---|
| # tokens | 1.8B˜ | 162M˜ | 24M˜ | 43M˜ | 33M˜ |
| ATTR | 41.5 | 39.9 | 44.3 | 36.7 | 39.4 |
| Overlap RT-Train | 83.7 | 61.6 | 73.3 | 70.9 | 68.6 |
| Overlap RT-Test | 87.5 | 69.9 | 76.7 | 75.3 | 74 |
| Overlap OpeNER-Train | 89.1 | 70.3 | 81.3 | 78.1 | 75 |
| Overlap OpeNER-Test | 96 | 89.1 | 92 | 90 | 90 |

Table 4: Average Type/Token ratio (ATTR) and overlap with test vocabulary in each training corpus.

## 5.2 Types of missing words

After seeing the number of words which were missing, we decided to study these qualitatively. We collected a list of the words that appear in the RT and OpeNER datasets which were not found in the embeddings.

A revision of each part of speech category revealed that the nouns not found in the embeddings models were often misspellings of more common nouns (*fantasi, alientation*) or creative use of language (*vidgame, dateflick, splatterfests, artsploitation, witlessness, drippiness, naturedness, diciness, gooeyness, baaaaaaaad*), both of which are important for categorization.

Even more important are the adjectives (*unrecommendable, unsuspenseful, uncinematic, unslick, unfakeable, snazzy*) and adverbs (*heartwarmingly, dullingly, repellently, forgettably, uncharismatically, appallingly*) which are not found in these models. These are words that often carry the most sentiment information in a sentence. The number of words lost due to filtering for subjectivity also seems to correlate with the performance of each model, as the subj-Europarl model lost the least number of wordforms and performed the best among the filtered corpora.

## 6 Experiments on under-resourced languages

In order to confirm our theory that this technique can improve representations for under-resourced languages, we perform the same concatenation experiment as in Section 4.2 but with Catalan data. The Catalan Wikipedia used in this experiment was from a wikipedia dump in January, 2016 and contains 182 million tokens. In order to gather task-specific data, we scraped 21,778 reviews in Catalan from `www.booking.com` which contain a total 4,161,719 tokens. We preprocessed both datasets in a similar fashion (lowercased, sentence tokenized, tokenized) using Freeling (Padró and Stanilovsky, 2012).

We test this on the Catalan Aspect-level Sentiment Dataset. This dataset was compiled from 879 Booking.com reviews collected in December of 2016. Three annotators annotated opinion holders, opinion targets, and four levels of sentiment (Strong Positive, Positive, Negative and Strong Negative). Statistics are shown in Table 5.

The results on the Catalan dataset (see Table 6) suggest that under-resourced languages benefit more from concatenating vectors trained separately than languages such as English. This is

|  | Training | Dev | Test |
|---|---|---|---|
| Num. phrases | 922 | 62 | 248 |
| Strong Pos | 27.3% | 25.8% | 27.4% |
| Pos | 34.2% | 33.9% | 34.3% |
| Neg | 33.3% | 33.9% | 33% |
| Strong Neg | 5.2% | 6.5% | 5.2% |

Table 5: Statistics of the Catalan Aspect-level Sentiment Dataset.

|  | 50 | 100 | 200 | 300 |
|---|---|---|---|---|
| Wikipedia | 44.9 | 61 | 72.2 | **75.4** |
| Booking | 55.61 | 62 | 72.7 | 69.5 |
| Appended | **61.4** | 60.4 | 72.2 | 70.6 |
| Spliced | 49.2 | 62 | 70.58 | **75.4** |
| Booking + Wiki | 59.4 | **62.6** | **73.3** | **75.4** |

Table 6: Macro average of classification scores on the Catalan Aspect-level Sentiment Dataset (4 class). **Bold** indicates the highest performance and underline indicates those that beat the generic baseline.

most likely due to the size and breadth of the English Wikipedia corpus. Therefore, for languages that do not have large, broad Wikipedia, we recommend training separate vectors and concatenating them afterwards.

## 7 Conclusion and Future Work

In this paper, we have explored several methods for improving sentiment analysis using word embeddings. First, we demonstrated that the subjectivity of a corpus is a viable method to determine the usefulness of a corpus for the task of sentiment analysis. We then show that concatenation of vectors trained on generic and task-specific datasets outperforms a single representation trained on both datasets. We also showed that it is possible to approximate task-specific data by extracting subjective portions from generic corpora. Finally, we showed that it is possible to use this technique to improve sentiment classification for under-resourced languages, even if there is little task-specific data available.

Instead of relying on subjectivity as metric, one could naturally use polarity to determine the usefulness of a passage for training embeddings for a sentiment related task. By removing neutral passages and keeping only those with positive or negative sentiment, it may be possible to achieve similar results to ours. We plan to explore this possi-

bility in the future.

We also believe that these techniques could improve results on tasks other than sentiment analysis, but we leave this for future work.

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
