# Peer review of "Improving sentiment classification with task-specific data"

_ACL 2017 — decision unknown_

[Official Review · Reviewer 1 · rating 2 · confidence 3]
soundness 5 · originality 5 · clarity 4 · impact 3 · substance 2 · appropriateness 5 · meaningful comparison 3 · presentation format Poster

This paper compares different ways of inducing embeddings for the task of
polarity classification. The authors focus on different types of corpora and
find that not necessarily the largest corpus provides the most appropriate
embeddings for their particular task but it is more effective to consider a
corpus (or subcorpus) in which a higher concentration of subjective content can
be found. The latter type of data are also referred to as "task-specific data".
Moreover, the authors compare different embeddings that combine information
from "task-specific" corpora and generic corpora. A combination outperforms
embeddings just drawn from a single corpus. This combination is not only
evaluated on English but also on a less resourced language (i.e. Catalan).

- Strengths:
The paper addresses an important aspect of sentiment analysis, namely how to
appropriately induce embeddings for training supervised classifers for polarity
classification. The paper is well-structured and well-written. The major claims
made by the authors are sufficiently supported by their experiments.

- Weaknesses:
The outcome of the experiments is very predictable. The methods that are
employed are very simple and ad-hoc. I found hardly any new idea in
that paper. Neither are there any significant lessons that the reader learns
about embeddings or sentiment analysis. The main idea (i.e. focusing on more
task-specific data for training more accurate embeddings) was already published
in the context of named-entity recognition by Joshi et al. (2015). The
additions made in this paper are very incremental in nature.

I find some of the experiments inconclusive as (apparently) no statistical
signficance testing between different classifiers has been carried out. In
Tables
2, 3 and 6, various classifier configurations produce very similar scores. In
such cases, only statistical signficance testing can really give a proper
indication whether these difference are meaningful. For instance, in Table 3 on
the left half reporting results on RT, one may wonder whether there is a
significant difference between "Wikipedia Baseline" and any of the
combinations. Furthermore, one doubts whether there is any signficant
difference between the different combinations (i.e. either using "subj-Wiki",
"subj-Multiun" or "subj-Europarl") in that table.
The improvement by focusing on subjective subsets is plausible in general.
However, I wonder whether in real life, in particular, a situation in which
resources are sparse this is very helpful. Doing a pre-selection with
OpinionFinder is some pre-processing step which will not be possible in most
languages other than English. There are no equivalent tools or fine-grained
datasets on which such functionality could be learnt. The fact that in the
experiments
for Catalan, this information is not considered proves that. 

Minor details:

- lines 329-334: The discussion of this dataset is confusing. I thought the
task is plain polarity classification but the authors here also refer to
"opinion holder" and "opinion targets". If these information are not relevant
to the experiments carried out in this paper, then they should not be mentioned
here.

- lines 431-437: The variation of "splicing" that the authors explain is not
very well motivated. First, why do we need this? In how far should this be more
effective than simple "appending"?

- lines 521-522: How is the subjective information isolated for these
configurations? I assume the authors here again employ OpinionFinder? However,
there is no explicit mention of this here.

- lines 580-588: The definitions of variables do not properly match the
formula (i.e. Equation 3). I do not find n_k in Equation 3.

- lines 689-695: Similar to lines 329-334 it is unclear what precise task is
carried out. Do the authors take opinion holders and targets in consideration?

***AFTER AUTHORS' RESPONSE***
Thank you very much for these clarifying remarks.
I do not follow your explanations regarding the incorporation of opinion
holders and targets, though.

Overall, I will not change my scores since I think that this work lacks
sufficient novelty (the things the authors raised in their response are just
insufficient to me). This submission is too incremental in nature.

[Official Review · Reviewer 2 · rating 3 · confidence 4]
soundness 5 · originality 5 · clarity 5 · impact 3 · substance 3 · appropriateness 5 · meaningful comparison 3 · presentation format Poster

- Strengths: An interesting and comprehensive study of the effect of using
special-domain corpora for training word embeddings.  Clear explanation of the
assumptions, contributions, methodology, and results.  Thorough evaluation of
various aspects of the proposal.

- Weaknesses: Some conclusions are not fully backed up by the numerical
results.  E.g., the authors claim that for Catalan, the improvements of using
specific corpora for training word vectors is more pronounced than English.  I
am not sure why this conclusion is made based on the results.  E.g., in Table
6, none of the combination methods outperform the baseline for the
300-dimension vectors.

- General Discussion: The paper presents a set of simple, yet interesting
experiments that suggest word vectors (here trained using the skip-gram method)
largely benefit from the use of relevant (in-domain) and subjective corpora. 
The paper answers important questions that are of benefit to practitioners of
natural language processing.  The paper is also very well-written, and very
clearly organized.